# Development of a Methodology for the Quality Management of Duplex Stainless Steels

**DOI:** 10.3390/ma15176008

**Published:** 2022-08-31

**Authors:** Aleksandr Fedorov, Andrey Zhitenev, Vladimir Karasev, Aleksey Alkhimenko, Pavel Kovalev

**Affiliations:** 1Scientific and Technical Complex “New Technologies and Materials”, Peter the Great St. Petersburg Polytechnic University, 195251 St. Petersburg, Russia; 2Institute of Machinery, Materials, and Transport, Peter the Great St. Petersburg Polytechnic University, 195251 St. Petersburg, Russia

**Keywords:** duplex stainless steels, quality control, quantification, thermodynamic database, modelling, austenite, δ-ferrite, σ-phase, non-metallic inclusions, etching

## Abstract

The use of traditional materials leads to failures and breakdowns of expensive equipment, so advanced materials are needed that can provide reliable and durable solutions. The ability to control the quality of duplex stainless steels (DSSs) can greatly help with the development of new compositions or choosing existing DSSs. In this case, it is necessary to consider the final consumer properties—corrosion resistance and mechanical properties, which depend on the phase composition, contamination with non-metallic inclusions (NMIs), and the presence of undesirable secondary phases. In this research, specimens of cast DSSs of different grades, produced at laboratory and industrial scales, were studied. A technique for quantifying the microstructure of DSSs was developed. A thermodynamic database was chosen that adequately describes the processes of phase formation in DSSs. The effects of heat treatment on the microstructure and corrosion properties of cast DSSs were studied. The effects of the structural state on the changes in consumer properties of the final product are shown. It is shown that using various deoxidation technologies, it is possible to obtain both NMIs that are dangerous in terms of corrosive activity and ones that are relatively safe.

## 1. Introduction

The oil, gas, and chemical industries are in need of advanced materials capable of providing reliable and durable solutions. The use of traditional structural steel leads to failures and breakdowns of expensive equipment [1,2]. The introduction of duplex stainless steels (DSSs) of various grades and cast products from them—such as various impellers, wheels, valves, and moving parts of equipment operating under aggressive conditions—can solve this problem.

DSSs are classified into grades according to PREN (pitting resistance equivalent number). The higher the PREN value, the more resistant the stainless steel is to localized pitting corrosion [3]. There are several empirical formulas to calculate the PREN (Equation (1) [4,5,6,7], Equation (2) [8], Equation (3) [3,4,7]):PREN = %Cr + 3.3(%Mo + 0.5%W) + 16%N;(1)
PREN = (%Cr − 14.5%C) + 3.3%Mo + 2%W + 2%Cu + 16%N;(2)
PREN = %Cr + 3.3%Mo + 16%N(3)

All other things being equal, the properties of DSSs depend on the ratio of ferrite and austenite [9,10,11,12,13], and on the content of secondary phases precipitation [14]. The control of the DSSs’ phase composition is a search for a compromise [15]. An increase in the chromium content in steel provides it with higher corrosion resistance, but to maintain the ferrite–austenite phase balance, it is necessary to increase the concentration of expensive nickel, which promotes the formation of the sigma (σ) phase [16], or nitrogen, which leads to the formation of unfavorable chromium nitrides [17]. Other secondary phases, such as χ, π, and R, can also affect the properties of DSSs, but the negative effects of the σ phase and chromium nitrides are predominant [18,19]. Therefore, it is necessary to solve the optimization problem of finding a composition that could provide an equal ferrite–austenite ratio at the heat treatment temperature of the cast products, while simultaneously ensuring that the temperature of the beginning of the formation of the σ phase and chromium nitrides is minimal.

When creating or choosing a particular DSS, it is necessary to consider the metallurgical history of the steel. Equations (1)–(3) consider only the elements that have a positive effect on the resistance to pitting corrosion, but they do not consider the harmful impurities of sulfur and oxygen [6], which form non-metallic inclusions (NMIs) that reduce the corrosion resistance of steels [20,21,22,23]. The distribution, quantity, size, and composition of NMIs depend on the technology of steel melting and deoxidation [24], and affect the corrosion resistance [25]. For example, pitting often occurs at the interface between inclusions and the metal matrix [25,26]. However, at present there is no unequivocal opinion on the contribution of certain NMIs to the formation of corrosion resistance [23,25]. Complex inclusions containing aluminum, magnesium, calcium, and silicon oxides, as well as spinel-like NMIs containing chromium, manganese, iron, and aluminum, are the most harmful NMIs [25]. Manganese sulfides have an unambiguously negative effect [20,21]. The neutral inclusions include magnesian spinel and titanium nitrides [25]. It is also believed that inclusions containing REMs increase corrosion resistance [23,26]. However, in several works there are contradictions about the effects of NMIs on pitting, so the nature of the effect, along with the search for links with metallurgical technology, requires an appropriate clarification.

Traditionally, thermodynamic modeling is used to solve these problems, together with the study of experimental specimens. However, for highly alloyed systems, there are problems of choosing thermodynamic databases, choosing a method for estimating the structure and, finally, solving the final problem of predicting properties, considering the metallurgical history of the steel. Therefore, the purpose of this research was to create a unified methodological approach to quality control of DSSs.

## 2. Materials and Methods

The experimental specimens of cast DSSs of different grades, produced at laboratory and industrial scales, were investigated (Table 1). The melting of laboratory steels (1–6) was carried out in an open induction furnace. Casting of laboratory steels was carried out at 1485 °C into copper molds and sand molds with a cross-section of up to 40 mm and a height of 100 mm. Industrial DSSs (7–8) were produced in open induction furnaces of various capacities from 100 to 500 kg. The experimental specimens were solution-annealed at 1050–1250 °C for 60 min and then quenched with water.

Metallographic studies were performed using optical microscopy methods on a Reichert-Jung MeF3A microscope (Reichert Inc., Depew, NY, USA) equipped with a Thixomet Pro [27] image analyzer (Thixomet Ltd., St.-Petersburg, Russia). The specimens were ground using 60–1200-grit abrasive paper and polished using 6–0.5 µm diamond suspensions on a Buehler EcoMet IV variable-speed grinder polisher (Buehler Ltd., Lake Bluff, IL, USA).

Thermodynamic modeling of phase-formation processes was carried out using the Thermo-Calc software (TCW5, Thermo-Calc Software Inc., Solna, Sweden) equipped with the TCFE [28] database and FactSage software (GTT-Technologies, Herzogenrath, Germany & Centre for Research in Computational Thermochemistry, Montréal, Canada) equipped with the 2007 and 2017 SGTE databases [29]. The chemical composition was determined using an optical emission spectrometer. Determination of the local chemical composition in individual structural components and non-metallic inclusions was carried out using a Zeiss Supra scanning electron microscope (SEM) (Carl Zeiss AG, Oberkochen, Germany) equipped with an energy-dispersive spectrometer. The phase content of the specimens was determined using a Bruker X-ray diffractometer (Bruker Ltd., Billerica, MA, USA).

Corrosion tests were carried out according to ASTM G48 (Method B) in an FeCl_3_ solution at 50 °C for 72 h. The size of the specimens was 20 mm × 20 mm × 3 mm. Sample surfaces were prepared on 120-grit abrasive paper. Before testing, the samples were cleaned with an alcohol-containing solution and acetone, and then cylindrical TFE-fluorocarbon blocks were fastened on them. After the test was completed, the samples were cleaned in a Buehler UltraMet V Ultrasonic Cleaner (Buehler Ltd., Lake Bluff, IL, USA) to remove corrosion products.

Electrochemical tests were carried out using a VersaStat Princeton Applied Research potentiostat (AMETEK Inc., Berwyn, PA, USA) at 22 °C in 5% NaCl solution with pH = 3 (acidified with acetic acid). All potentials were referred to the silver chloride reference electrode (Ag/AgCl). The pitting potential E_pit_ was determined in accordance with ISO 17475:2005. The linear anodic polarization was carried out at a scan rate of 0.16 mV/s. For each experiment, the duration of the corrosion potential (or open-circuit potential) was 2–5 mV. The size of the specimens was 20 mm × 20 mm. The non-tested parts of each specimen were isolated with a wax–resin coating so that the area of the test surface was 1 cm^2^. Before testing, the surface of the samples was cleaned with alcohol-containing solution and acetone.

Macrohardness to assess the strength of the experimental steels was measured using the Zwick/Roell ZHU 8187.5 (ZwickRoell GmbH & Co. KG, Ulm, Germany) using the Vickers method with a load of 10 kg and a holding time of 10 s.

## 3. Results

To solve the main task of this paper, the secondary tasks were successively solved. Therefore, first it was considered the limitations of existing tools for modeling the structures and studying duplex steels; then, based on this, it was developed new and improved versions of existing methods; after that, it was solved the problem of finding the ideal DSS composition, and then it was made experimental adjustments related to the features of the real technology and the formation of secondary phases. Finally, it was assessed the impact of metallurgical technology on quality. An independent section of this paper is devoted to each problem, but each takes into account all of the previous results.

### 3.1. Development of Tools to Ensure the Quality of Duplex Stainless Steels

Today, traditional tools for the development of alloys, which are also used to improve the quality of DSSs, include thermodynamic modeling [30], which has been successfully used for many applications, along with the use of various quantitative methods for structural evaluations to compare simulated and experimental results [31]. We considered the applicability of these methods to the experimental steels.

There are many thermodynamic databases optimized for different systems [28,29]. This gives rise to uncertainty in the choice of initial conditions for the calculation in each specific case, since it is the information support of mathematical modeling that determines the result obtained. This is shown in Figure 1, which shows the results of calculating the behavior of δ-ferrite for steel with 0.02%C-26%Cr-0.6%Si-1.6%Mn-6%Ni-0.5%Mo-0.04%N-0.17%Cu (Steel 3, Table 1).

Depending on the choice of database, the predicted phase composition is very different. Hence, when calculating using the SGTE 6.0 and SGTE 7.3 databases, the type of the state diagram changes significantly compared to the calculation using the TCFE database. It is estimated that 80%, 95%, and 100% of δ-ferrite is released during solidification, as calculated with SGTE 6.0, SGTE 7.3, and TCFE, respectively. Another significant difference is that, according to TCFE, there is a δ-ferrite stability interval extending to 1210 °C for the considered steel composition (solid line), while according to SGTE, only 80% (SGTE 7.3) and 65% of δ-ferrite (SGTE 6.0) exists at this temperature. Figure 1 shows a line corresponding to an equal ratio of δ-ferrite and austenite (gray line)—that is, the required phase balance for DSSs—and the temperature at which this ratio can be achieved differs by almost 200 °C, depending on the chosen thermodynamic database.

An adequate choice of database for the calculation is possible only if the experimental results obtained under fully controlled conditions are correctly compared directly with the calculation results. There are physical methods of quantitative assessment, e.g., X-ray diffraction (XRD) quantitative analysis, or magnetic methods [32]. However, these require reference databases to interpret the results, and the results depend on how the specimens are prepared. The simplest and most effective method is to use an automatic image analyzer, but this requires obtaining a phase-contrast image. Various authors suggest a variety of etchants [31], but a later analysis [30] carried out by our team showed that even the use of electrolytic etching (Figure 2a) does not allow sufficient phase contrast to be obtained for automatic analysis.

Beraha (20 mL of HCl, 80 mL of water, and 1 g of K_2_S_2_O_5_) is the most effective etchant for DSSs [30], which makes it possible to reveal the precipitation of both the main phases—austenite and ferrite (Figure 2b)—and the secondary phases, i.e., the sigma phase (Figure 2c), chromium nitrides, and Laves phases (Figure 2d). The images obtained after etching with Beraha are quite contrasting, and allow us to evaluate all phases in DSSs [30], while also making it possible to estimate the volume content, average and maximum sizes, distribution density, and other metric parameters. The images obtained using Beraha are suitable for the application of automatic analysis methods provided, for example, in ASTM E 1245, which can subsequently be integrated into the quality management system during the production of DSSs.

Test measurements were carried out on the experimental Steels 1–3, differing only in chromium content, after a series of solution annealing in the temperature range of 1050–1250 °C to obtain a different phase balance, with all other things being equal. 

Comparison of the results of thermodynamic calculations and experimental results obtained using the developed method of quantitative assessment made it possible to choose a thermodynamic database that most correctly describes the processes (Figure 3a). The results obtained were also compared with the results of XRD with respect to the results of thermodynamic modeling (Figure 3b). Even though the obtained straight lines were quite close, the scatter of the XRD results was large compared to the results of automatic image analysis after selective etching with Beraha etchant.

The obtained experimental data made it possible to determine the thermodynamic database that most correctly describes the real structure (Figure 3b). Calculations using the SGTE 6.0 database describe the phase balance of experimental specimens the worst. The SGTE 7.3 database can significantly improve the accuracy of the calculation. However, only the calculation using the TCFE database makes it possible to predict the phase composition of DSSs almost perfectly.

Thus, the developed method of quantitative assessment made it possible to verify the results of thermodynamic modeling on laboratory and industrial samples to select the most correct model.

### 3.2. Effect of Chemical Composition on Phase Balance

By choosing an adequate thermodynamic database for the calculations, it becomes possible to analyze the behavior of the phase composition depending on the chemical composition. As follows from the definition of DSSs, austenite and δ-ferrite after heat treatment should be in an equal ratio, while there should be the absence or low-temperature onset of the formation of secondary phases. The requirements for the chemical composition of these steels can be formulated as follows: the temperature at which austenite and ferrite are in equal proportions $T_{50/50}^{\gamma /\delta }$ should be in the range of the possible heat treatment temperature of the casting, and the temperature at which the σ phase begins to form $T_{0}^{\sigma }$ and chromium nitrides $T_{0}^{Cr2N}$ should be as low as possible in this range to exclude the presence of these phases in the steel structure.

It should be noted that a similar approach has already been used in a previous paper, but only with the use of databases of thermodynamic data from SGTE [8]. Therefore, in this paper, these results were refined using more advanced thermodynamic databases.

As shown below by the analysis of the phase-formation processes in two DSSs of different compositions (Figure 4), such a requirement entails a search for a compromise. If the temperature $T_{50/50}^{\gamma /\delta }$ at which austenite and ferrite are in equal proportions is in the temperature range of the heat treatment, then the temperature of the beginning of the formation of the σ phase $T_{0}^{\sigma }$ turns out to be too high (Figure 4a). If the composition is chosen so that the σ phase begins to form at low temperatures, then the point $T_{50/50}^{\gamma /\delta }$ becomes too high (Figure 4b)—that is, the development and solution of the problem of optimizing the chemical composition is required.

Consider the influence of the base elements (Cr and Ni) on the position of temperatures $T_{50/50}^{\gamma /\delta }$ and $T_{0}^{\sigma }$ relative to the temperature range of solution annealing. With an increase in the chromium content in the DSSs’ composition at a constant nickel content, the temperature $T_{50/50}^{\gamma /\delta }$ decreases significantly, while $T_{0}^{\sigma }$, on the other hand, increases (Figure 5).

These graphs show the values of the criteria $T_{50/50}^{\gamma /\delta }$ (Figure 5a) and $T_{0}^{\sigma }$ (Figure 5b), calculated for common commercial DSS [11] at the limiting concentrations of alloying elements within the grades of these steels. As follows from the analysis of these results (Figure 5), not all commercial DSSs satisfy the developed criteria $T_{50/50}^{\gamma /\delta }$ and $T_{0}^{\sigma }$.

Figure 5c,d show how critical DSSs’ temperatures can be controlled by adding other elements. If 1.5% molybdenum is added to Steel 1 (red line), then the temperature $T_{50/50}^{\gamma /\delta }$ will decrease significantly. When 4% manganese is added to the same steel, on the other hand, an equal amount of austenite and ferrite is achieved at higher temperatures. At the same time, molybdenum and manganese additives increase the temperature at which the sigma phase begins to form. Using Steel 2 as an example (the blue line in Figure 5c,d), we considered the effects of nitrogen and copper, which significantly increase the temperature $T_{50/50}^{\gamma /\delta }$ and practically do not affect the temperature at which the sigma phase begins to form.

Thus, by varying the alloy contents, it is possible to purposefully change $T_{50/50}^{\gamma /\delta }$ and $T_{0}^{\sigma }$, ensuring their optimal values. In steels alloyed with nitrogen, along with these criteria, the temperature at the beginning of the formation of $T_{0}^{Cr2N}$ should be considered.

To take into account the effects of all elements that make up common DSSs, more than 400 calculations of various variants of DSS compositions were carried out using thermodynamic modeling methods to cover all ranges of changes in the compositions (wt.%): C—0.03…0.08; Cr—19…31; Mn—1…10; Ni—4.5…10; Mo—1…5; V—0…1.5; N—0.01…0.6; Si—0.1…1; Cu—0...1.25; Ti—0…0.7; W—0…3; Nb—0…2. As a result of these calculations, we obtained a database of the chemical compositions of steels and the corresponding temperatures of the coexistence of ferrite and austenite in equal shares $T_{50/50}^{\gamma /\delta }$, along with the temperature of the beginning of the formation of the σ phase ($T_{0}^{\sigma }$) and the formation of $T_{0}^{Cr2N}$. The results of the $T_{50/50}^{\gamma /\delta }$, $T_{0}^{\sigma }$, and $T_{0}^{Cr2N}$ calculations for all studied DSSs were summarized using regression analysis carried out using Statistica software. Regression equations were obtained that adequately describe the effects of the chemical composition of steels on the criteria $T_{50/50}^{\gamma /\delta }$, $T_{0}^{\sigma }$, and $T_{0}^{Cr2N}$:(4)$$T_{50/50}^{\gamma /\delta }=1730+1148\times \%C-44\times \%\mathrm{Cr}+12\times \%\mathrm{Mn}+55\times \%\mathrm{Ni}-45\times \%\mathrm{Mo}+972\times \%N-63\times \%\mathrm{Si}+75\times \%\mathrm{Cu}$$
(5)$$T_{0}^{\sigma }=594+44\times \%C+5\times \%\mathrm{Cr}-4\times \%\mathrm{Mn}+11\times \%\mathrm{Ni}+41\times \%\mathrm{Mo}-18\times \%N-3\times \%\mathrm{Si}+7\times \%\mathrm{Cu}$$
(6)$$T_{0}^{Cr2N}=557+2510\times \%C+6\times \%\mathrm{Cr}-11\times \%\mathrm{Mn}-2\times \%\mathrm{Ni}+13\times \%\mathrm{Mo}+692\times \%N+35\times \%\mathrm{Si}+7\times \%\mathrm{Cu}$$

Figure 6 shows the scatterplot for temperatures $T_{50/50}^{\gamma /\delta }$, $T_{0}^{\sigma }$, and $T_{0}^{Cr2N}$ calculated from these equations on the one hand, and temperatures obtained from thermodynamic simulations on the other.

This diagram illustrates the high adequacy of the obtained regression equations—the coefficients of determination of straight lines 1:1 exceed 0.99.

The resulting equations describing the behavior of the developed criteria based on the TCFE database allow us to quickly determine the chemical composition and processing parameters of steels. However, the actual properties depend on many factors, which we consider below.

### 3.3. Effects of Chemical and Phase Composition on Properties

Traditionally, in the practice of researchers and representatives of industries, the search for the most effective technology is carried out in a rather simple way, where they enumerate various technological modes and monitor the relationship between changes in parameters—for example, heat treatment—and the final properties of the product. However, this approach does not always make it possible to identify unambiguous dependencies, and often leads to an impasse. Figure 7a shows an example of the analysis of the effect of quenching temperature on hardness. There is a general trend towards an increase in hardness with an increase in the heating temperature; however, it is difficult to interpret the results obtained. Using the developed approaches to the study of DSSs, it becomes possible to obtain adequate dependencies showing the relationship between the structure and properties of the final product. Figure 7b shows the same data, but already processed using the developed criteria. It follows from the results that it is the increase in the proportion of delta-ferrite and a significant deviation from the optimal phase balance that lead to an increase in hardness and, hence, strength, and to a regular decrease in plasticity. Thus, the deviation from the optimal values of the previously selected criteria significantly affects the changes in the mechanical properties of DSSs.

In terms of corrosion properties, the situation is more complex. Figure 8a,c show the dependences of the pitting potential (*E*_pit_) and the crevice corrosion rate (CR), respectively, on the quenching temperature. The values of *E*_pit_ and CR do not depend on the quenching temperature, and this does not allow one to determine the optimal heat treatment regime. However, by rearranging these dependences on the content of delta-ferrite, predicted by thermodynamic modeling at given temperatures, it is possible to assess what exactly has a critical influence on the formation of the properties of the final product. First, for steels of different grades, depending on PREN, the absolute values of the CR and the *E*_pit_ are different. The behavior of these characteristics in different steels also differs. For the experimental Steels 1–3 with PREN 24–29, the CR (Figure 8d) slightly increases with distance from the optimal phase composition, since crevice corrosion processes are less sensitive to structural changes. However, in these steels, there is a clear maximum of the pitting potential *E*_pit_ at 65% ferrite (Figure 8b). Super duplex Steel 8 with PREN 38 has no microalloying and small content of nitrogen, so there are no secondary phases [30]. The CR values increase with distance from the optimal phase composition. The CR values of Steel 7 with PREN 44 are maximal at a δ-ferrite content of 65%, and then the corrosion rate drops rapidly with an increase in the δ-ferrite content to 80%. This steel is alloyed with 0.1% Nb, so carbonitrides and Laves phases [33] are formed in it. Their contents decrease as the quenching temperature increases; therefore, despite the departure from the optimal phase composition, the properties increase. The behavior of *E*_pit_ is also determined by the presence or absence of secondary phases [30] in the super DSSs.

### 3.4. Effect of Non-Metallic Inclusions (NMIs)

Ensuring high PREN values and choosing the optimal heat treatment regime is no guarantee of obtaining good properties, as an unfavorable metallurgical history—in particular, suboptimal melting and deoxidation technologies—can have a detrimental effect. To minimize this impact, it is necessary to control deoxidation processes to control the composition of NMIs. The most used deoxidizers for stainless steels are titanium, aluminum, and rare-earth metals (REMs).

Let us show how different types of NMIs affect the corrosion properties of DSSs using the example of the experimental Steels 4–6 of the super duplex grades. These steels were examined after quenching from 1100 °C in the as-cast state.

The technique for studying the nucleation of pitting on NMIs was as follows: polishing the specimen and fixing the area of study; conducting potentiodynamic tests on a polished specimen; and studying the same plane after potentiodynamic tests.

Evaluation of the volume fraction of NMIs using an optical microscope showed that steel deoxidized with titanium is the most contaminated with inclusions (volume fraction of 0.021%). In steel deoxidized with aluminum, there are fewer inclusions (volume fraction of 0.014%), while in steel deoxidized with REMs, the volume fraction is the same (volume fraction of 0.015%), but their size is smaller. On Figure 9 shows the morphology of NMIs after deoxidation with various elements.

Such a change in the volume fraction is caused by a change in the type of inclusions when various deoxidizers are added. When Ti is added, large irregular-shaped inclusions are formed, consisting of titanium oxides and nitrides. When Al is added, they are transformed into dendritic inclusions of aluminum oxide. With the addition of active rare-earth metals (REMs), almost all inclusions are transformed into globular ones, which are easily removed from the steel melt. This is confirmed by the results of determining the chemical composition of the inclusions. The chemical composition of non-metallic inclusions affects not only corrosion resistance, but also other properties. Thus, the coefficient of thermal expansion (CTE) and the hardness of inclusions determine the occurrence of cavities and stresses between the inclusion and the steel matrix [34]. The difference in chemical potential predetermines the course of the anode or cathode process. The number and size of emerging micro-galvanic couples, the number of voids, etc., depend on the total number of inclusions, as well as their size and morphology. It should be noted that in steels deoxidized with Ti and Al, accumulations of inclusions are found, which are not found in steel deoxidized with REMs. These accumulations are represented by oxides and nitrides of Ti in Steel 4, and by pure corundum in Steel 5. Their formation occurs during the introduction of the deoxidizer when strong supersaturations occur at the site of the deoxidizer dissolution. REM additives completely modify such accumulations and transform them into rounded NMIs. The chemical composition of the NMIs is presented in Table 2.

The effects of deoxidation technology and NMI type on corrosion properties were assessed by determining the *E*_pit_ on the polarization curves. The results of the potentiodynamic tests are shown in Table 3 and Figure 10.

Since the *E*_pit_ of all specimens is approximately the same, Δ*E* was calculated. The larger the Δ*E* value, the better the pitting corrosion resistance of the material. It can be seen from the polarization curve that the resistance to pitting corrosion of steels deoxidized with Ti and REMs was approximately the same; however, the steel deoxidized with Al showed the worst results.

Ti (N, O) and (Ce, La, Al) O-type inclusions were not completely dissolved, while almost all Al_2_O_3_-type inclusions were dissolved (Figure 11). From the results of these experiments, it can be concluded that pitting most likely occurs on inclusions of the Al_2_O_3_ type.

### 3.5. Optimization of Chemical Compositions

The final stage in the development of ideas about the quality management of DSSs is the creation of a tool for predicting the optimal composition and production technology. Using Equations (4)–(6), we can solve the inverse problem of choosing the chemical composition of DSSs or the problem of optimizing existing grades that meet the given criteria (Table 2). Let us show how the resulting multiple regression equations can be used to optimize the composition of common DSSs and develop new compositions of DSSs.

Let us supplement the criteria proposed above, which determine the mechanical and technological properties, with the PREN criterion, which characterizes the resistance of steel to pitting corrosion and is calculated from its chemical composition. 

Optimization was carried out using the “Solver” of the MS Excel add-in program, with the help of which the steel composition was calculated from Equations (4)–(6), considering one of the criteria as the objective function and the specified restrictions on the remaining criteria. 

We selected the optimal compositions of the previously considered commercial grades, choosing PREN (4) as the objective function; then, for the remaining criteria, we introduced the following restrictions:(7)$$1050\;{}^{\circ}C\;<T_{50/50}^{\gamma /\delta }+\;\Delta T_{1}<\;1110\;{}^{\circ}C$$
(8)$$T_{0}^{\sigma }<\;1050-\Delta T_{1}\;$$
(9)$$T_{0}^{Cr2N}<\;T_{50/50}^{\gamma /\delta }-\Delta T_{2}$$

Here, in Condition (7), 1050 °C and 1110 °C are the recommended heat treatment intervals. The maximum temperature of the beginning of the formation of the sigma phase $T_{0}^{\sigma }$ must be lower than the heat treatment interval by the value Δ*T*_1_, considering the time of movement of the workpiece from the heating furnace to the quenching tank. The temperature of the beginning of the formation of chromium nitrides $T_{0}^{Cr2N}$ must be below the temperature $T_{50/50}^{\gamma /\delta }$, so as to exclude their formation during isothermal holding.

Consider the problem of developing new steels with specified technological and operational properties. Based on the analysis of common standards and publications [7,11], we selected the ranges of possible changes in the alloying elements of steels (Table 4). The optimization carried out showed that in some steels—for example, S32001 and S31200—it is possible to find the optimal chemical composition that satisfies the given conditions. However, in steel S32750, the optimal composition could not be found.

Thus, the practical value of the developed criteria for optimizing existing compositions of commercial DSSs and developing new DSS compositions with the desired properties is shown.

## 4. Conclusions

Thus, in this paper, tools for controlling the quality of DSSs at all stages of production—from melting, through considering the deoxidation technology, to final heat treatment, considering secondary phases—were developed.

A technique for quantitative assessment of the microstructure of DSSs based on etching with Beraha etchant and subsequent automatic analysis of the content of ferrite, austenite, and secondary phase precipitation using the ASTM E1245 method was developed. The results of thermodynamic modeling carried out on various thermodynamic databases were compared with the results of a quantitative assessment of the microstructure. The thermodynamic database that adequately described the processes of phase formation in DSSs was chosen.

The effects of heat treatment on the structure and corrosion properties of cast DSSs were studied. The influence of the structural state on the change in the pitting corrosion potential was shown. For a reasonable choice of the compositions of cast DSSs, thermodynamic criteria were developed. The behavior of the developed criteria with changes in the chemical composition of steels was studied, and the results of these studies were generalized in the form of multiple regression equations, which can be used in the development of new compositions and optimization of existing compositions of DSSs.

The effects of the deoxidation technology and the types of NMIs formed on the corrosion resistance of DSSs were studied. It was shown that using various deoxidation technologies, it is possible to obtain both Al_2_O_3_ NMIs, which are dangerous from the point of view of corrosivity, and relatively safe Ti (N, O), (Ce, La, Al) O, which are not initiators of pitting. It is not recommended to use Al as a deoxidizer in DSSs, since pitting is most likely to occur on inclusions of the Al_2_O_3_ type; these inclusions were completely dissolved during electrochemical tests. It is recommended to preliminarily deoxidize DSSs with Ti with the obligatory subsequent modification of REMs, so the introduction of these elements reduces the volume fraction and size of NMIs and converts them into an inert form.

## Figures and Tables

**Figure 1 materials-15-06008-f001:**
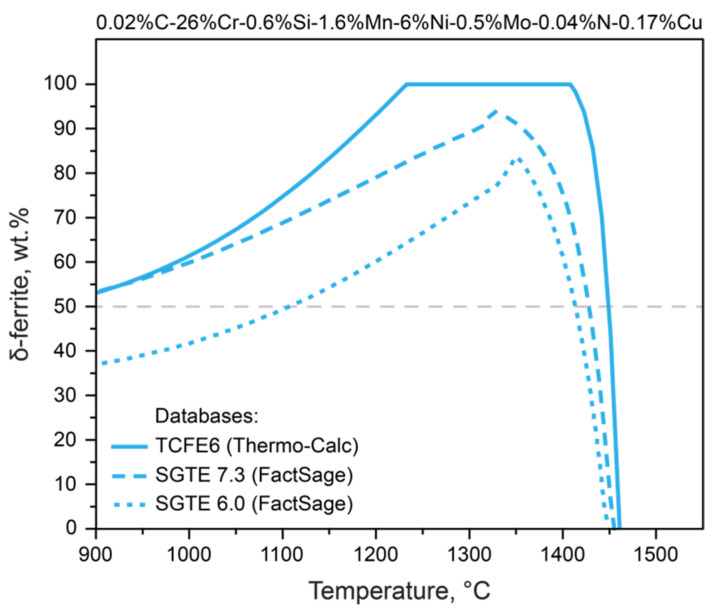
Thermodynamic modeling of δ-ferrite formation in Steel 3, performed using three thermodynamic databases.

**Figure 2 materials-15-06008-f002:**
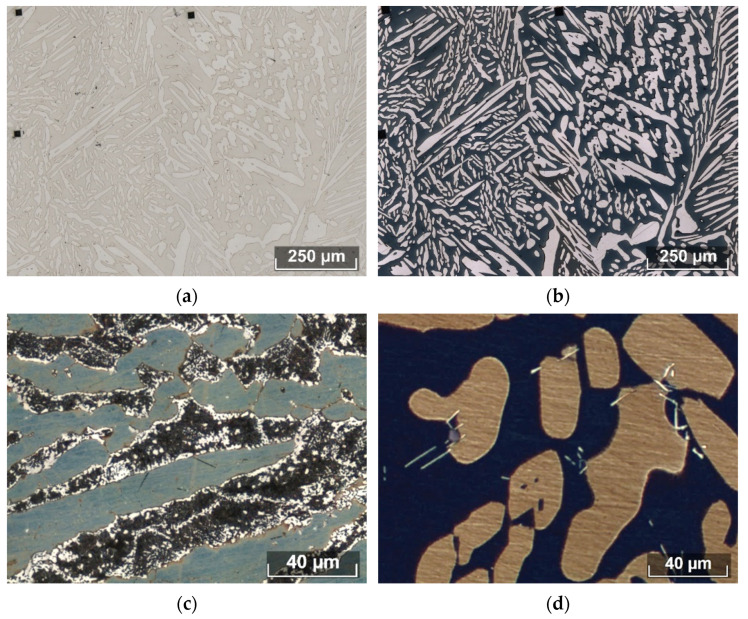
Microstructure of studied steels after electrolytic etching using NaOH (**a**) and chemical etching using Beraha etchant (**b**) in Steel 3. The Beraha etchant allows identification of the σ phase in Steel 7 (**c**) and Laves phases in Steel 8 (**d**).

**Figure 3 materials-15-06008-f003:**
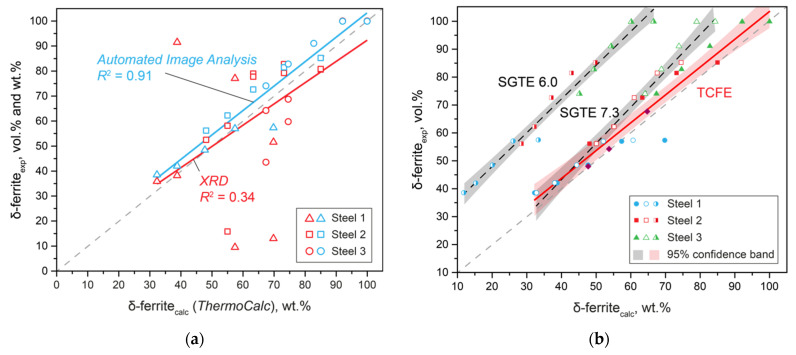
Comparison of phase volume fraction estimation methods (**a**), and selection of the adequate thermodynamic database (**b**).

**Figure 4 materials-15-06008-f004:**
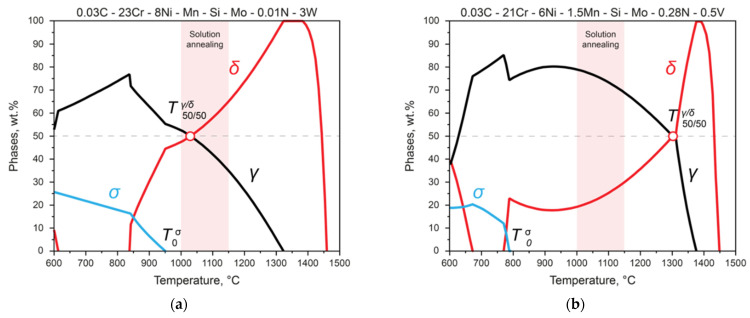
The temperature of δ-ferrite and austenite’s coexistence in equal fractions ($T_{50/50}^{\gamma /\delta }$), and the temperature at which σ-phase formation begins ($T_{0}^{\sigma }$ ), at two different chemical compositions of steels: (**a**) $T_{50/50}^{\gamma /\delta }$ ) is in the temperature range of the heat treatment, and $T_{0}^{\sigma }$ turns out to be too high; (**b**) $T_{50/50}^{\gamma /\delta }$ becomes too high, and $T_{0}^{\sigma }$ is in the low temperature region.

**Figure 5 materials-15-06008-f005:**
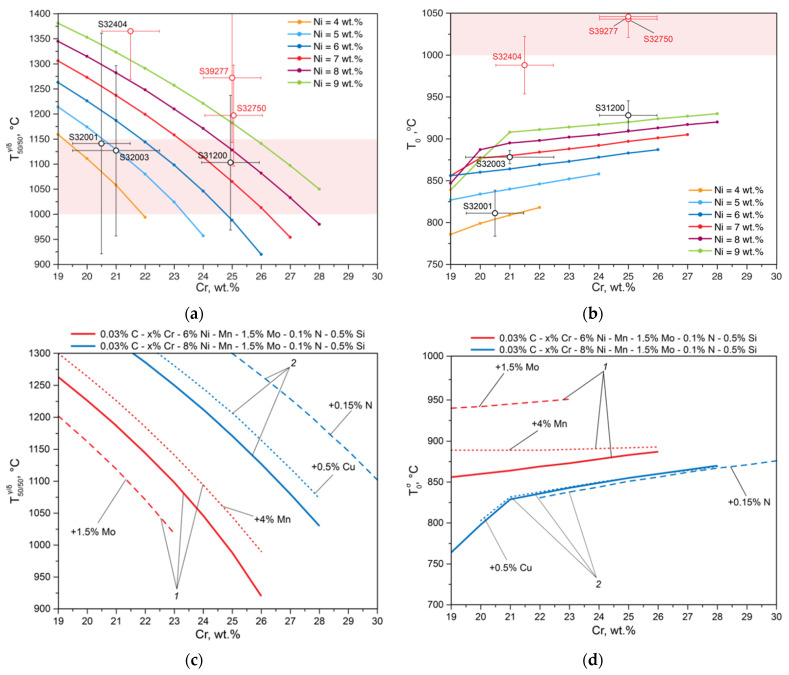
The effects of Cr and Ni content (**a**,**b**) and secondary alloying elements’ contents (**c**,**d**) in DSSs at temperatures $T_{50/50}^{\gamma /\delta }$ and $T_{0}^{\sigma }$.

**Figure 6 materials-15-06008-f006:**
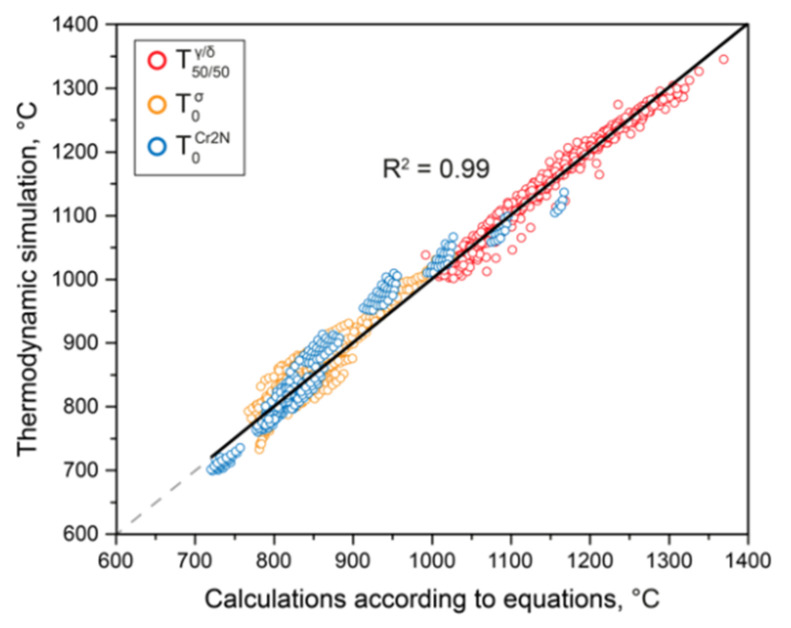
Scattering diagram for temperatures $T_{50/50}^{\gamma /\delta }$, $T_{0}^{\sigma }$, and $T_{0}^{Cr2N}$ calculated by Thermo-Calc with the TCFE database vs. prediction by Equations (4)–(6).

**Figure 7 materials-15-06008-f007:**
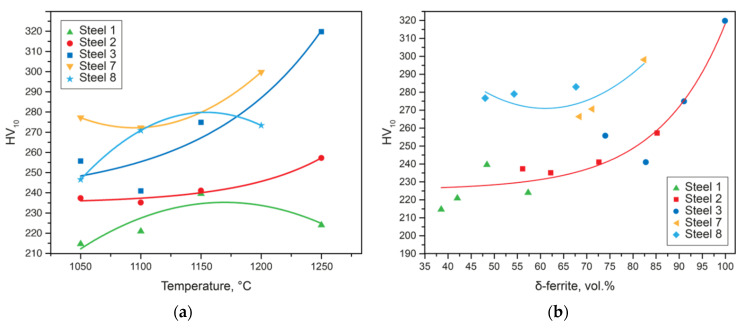
Correlation of hardness (HV) and the quenching temperature (**a**) and the amount of δ-ferrite in the steel microstructure (**b**).

**Figure 8 materials-15-06008-f008:**
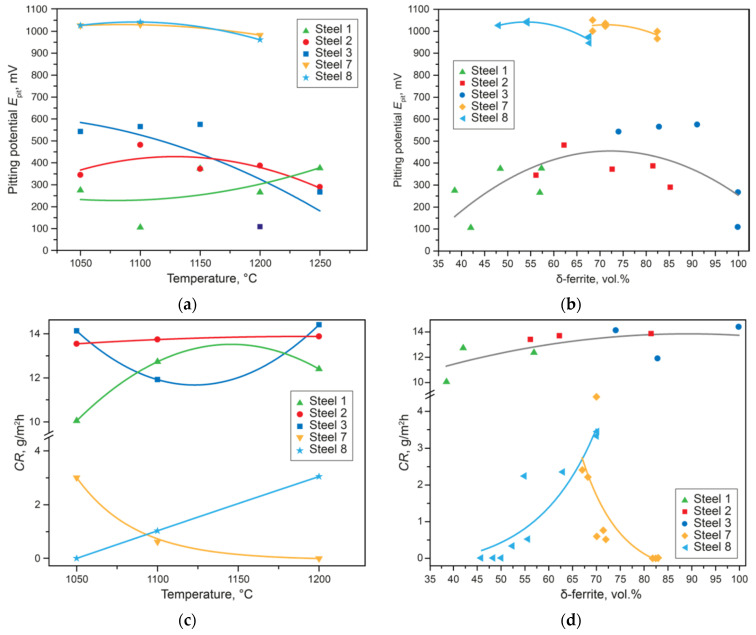
Quenching temperature’s effect on the pitting potential (**a**) and the crevice corrosion rate (**c**). Effects of the amount of δ-ferrite on the pitting potential (**b**) and the crevice corrosion rate (**d**).

**Figure 9 materials-15-06008-f009:**
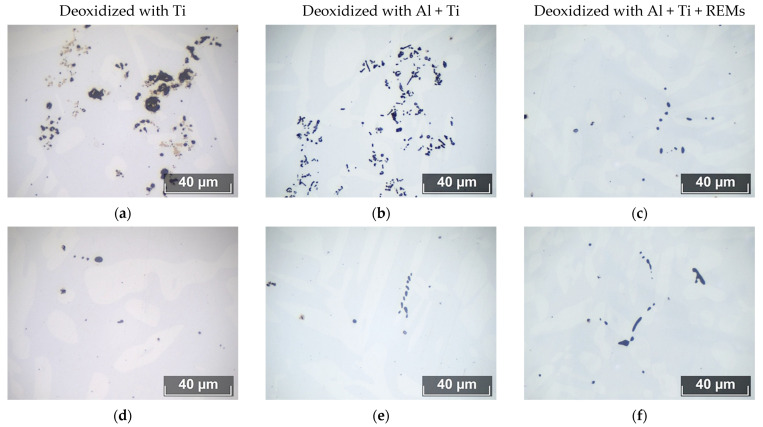
Sizes and morphologies of NMIs found in experimental Steels 4–6 with different deoxidation technologies. (**a**,**d**) NMIs after Ti is added; (**b**,**e**) NMIs after Al + Ti is added; (**c**,**f**) NMIs after Al + Ti + REMs is added.

**Figure 10 materials-15-06008-f010:**
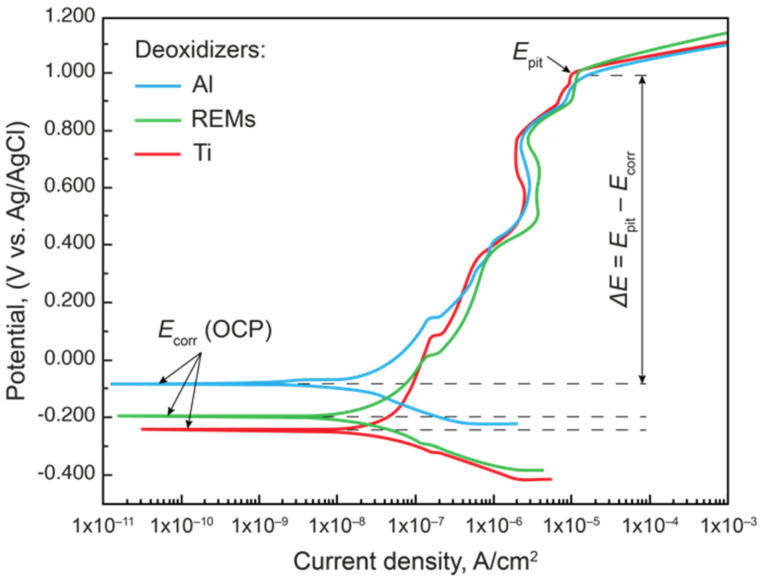
Polarization curves obtained on specimens of experimental Steels 4–6 with various NMIs obtained using different deoxidation technologies.

**Figure 11 materials-15-06008-f011:**
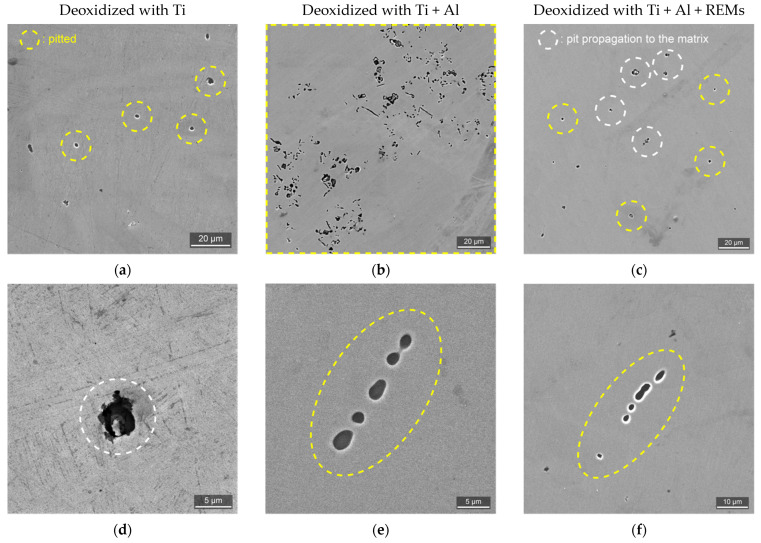
SEM images of the non-metallic inclusions in the experimental steels after the potentiodynamic tests: (**a**,**d**) NMIs after Ti is added; (**b**,**e**) NMIs after Al + Ti is added; (**c**,**f**) NMIs after Al + Ti + REMs is added.

**Table 1 materials-15-06008-t001:** Chemical compositions of experimental steels.

Steel	Element, wt.%												PREN **
	C	Cr	Si	Mn	Ni	Mo	N	Cu	Nb	Ti	Al	REMs *	
1	0.02	21.0	0.6	1.6	6.0	0.5	0.04	0.2	0.06	–	–	–	24
2		23.0								–	–	–	26
3		26.0								–	–	–	29
4	0.04	27.0	0.7	0.1	7.5	4.2	0.10	0.6	–	0.03	–	–	43
5											0.03	–	
6												0.1	
7	0.03	23.0	0.5	1.0	6.0	4.0	0.20	2.5	0.10	–	–	–	44
8	0.07	27.0	0.6	0.5	8.1	2.9	0.12	0.5	0.03	–	–	–	38

* Rare-earth metals. ** Calculations according to Equation (2).

**Table 2 materials-15-06008-t002:** Chemical composition of non-metallic inclusions after deoxidation with Ti, Al, and REMs.

[R]	Element, wt.%						
Ti	N	O	Al	Ti	S	La	Ce
	21.3	7.7	1.0	70.0	–	–	–
	–	53.0	8.5	38.5	–	–	–
	17.8	19.7	2.0	60.5	–	–	–
	23.0	5.3	0.3	71.4	–	–	–
Al		44.8	54.9	0.3	–	–	–
		42.9	56.8	0.3	–	–	–
		43.9	55.2	0.9	–	–	–
		42.9	53.4	3.7	–	–	–
REMs	–	28.2	19.3	4.3	1.1	6.7	40.4
	–	33.9	21.2	9.1	–	6.9	28.9
	3.3	42.1	25.0	18.8	1.3	1.8	7.7
	3.8	42.1	25.5	22.4	–	–	6.2

**Table 3 materials-15-06008-t003:** Results of the potentiodynamic tests.

[R]	Corrosion Potential *E*_corr_, (mV vs. Ag/AgCl)	Pitting Potential *E*_pit_, (mV vs. Ag/AgCl)	∆*E* = *E*_pit_ − *E*_corr_, mV
Ti	–248	1028	1276
Al	–86	1012	1098
REMs	–197	1039	1236

**Table 4 materials-15-06008-t004:** The search for steel compositions with the desired properties.

Grade	PREN	Criterion			Element, wt. %					
		$T_{50/50}^{\gamma /\delta }$	$T_{0}^{\sigma }$	$T_{0}^{Cr2N}$	Cr	Mn	Ni	Mo	N	Cu
	**Optimization completed successfully:**									
S32001	26.0	1050	741	742	21.5	4.0	2.0	0.6	0.05	1.0
S31200	34.8	1050	860	888	25.5	2.0	5.7	2.0	0.19	-
	**No optimal compositions among the given conditions:**									
S32750	40.0	1110	913	948	25.2	1.2	6.0	3.1	0.24	0.5

## Data Availability

Not applicable.

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
