# Peer review of "Development of a Methodology for the Quality Management of Duplex Stainless Steels"

_materials, 2022, doi:10.3390/ma15176008_

Round 1

Reviewer 1 Report

This study developed a technique for quantitative assessment of the microstructure of DSSs based on etching with Beraha etchant, and the automatic analysis of the content of ferrite, austenite, and secondary phase precipitation using the ASTM E1245 method was performed. The results of thermodynamic modeling carried out on various thermodynamic databases were compared with the results of a quantitative assessment of the microstructure. The effect of heat treatment on the structure and corrosion properties of cast DSSs was studied. The key points and innovation should be highlighted in this studied. Also, the description of the research work needs to be modified to make the key points stand out. Some improvements and explanations are needed for this paper when considering publication in the Journal.

1. The authors sate in Line 72-75 that “Casting of laboratory steels was carried out at 1485 °C into a copper mold and sand molds with a cross section of up to 40 mm and a height of 100 mm”, photographs of the cast materials or specimens in this study should be provided.

2. Photos of the devices used for the metallographic studies in Line 83-97 should be added.

3. The explanations of “TCFE”, “SGTE 7.3”, and “SGTE 7.3” shown in Fig. 1 should be added.

4. What are the differences between test results marked in red and blue in Fig. 3? What do the dark areas adjacent to the lines mean in Fig. 3(b)?

5. How did the authors get the data in Fig. 5 since the steels cannot be found in Table 1?

6. Fig. 10 shows the polarization curves obtained on experimental Steels 4-6 specimens, how to distinguish the results for different type of materials?

Author Response

Thank you for your comments!

Reviewer 2 Report

The study entitled "Development of the Methodology for Quality Management of 2 Duplex Stainless Steels" is quite interesting. However, the authors have generated a significant amount of data that has been poorly discussed. The poor discussion presented may lead readers to misinterpretations. I recommend that the authors do a solid review of this work before publication.

My remarks:

1. Introduction:

"The use of traditional materials leads to 26 failures and breakdowns of expensive equipment" - What are these materials? Why are they used instead of DSSs? For all DSSs are they always the best choice? What is the cost-benefit ratio?

How important is each element described in the PREN calculation equation? Readers do not always know the role of each element.

2. Materials and Methods 

Which reactive element was used for cleaning the DSS?

If we are talking about impurities that affect the pitting corrosion of the material, why Mn that exhibits a deleterious effect on pitting corrosion was not considered in the calculations? Do the PREN equations shown serve to calculate the PREN value of DSS with 1.6 wt.% Mn?

How were the XRD analyses performed? Why does the volume fraction estimated through the technique present a high dispersion in the results [Figure 3 (a)]?

How were the electrochemical tests performed? How was the experimental apparatus? how long of OCP? what was the scan rate? What was the criterion used to estimate the pitting potential?

Why was 10 kgf used to take hardness measurements?

3. Results

Are the authors sure that the Beraha reagent is the most indicated to reveal intermetallic phases? How was the electrolytic attack mentioned in the text carried out?

Which phase in DSS has higher mechanical strength, ferrite or austenite? Why does the hardness increase with the ferrite fraction? This result must be better explained.

The corrosion results require clarification:

Although there is an overall PREN value, the PREN value of ferrite differs from austenite due to the solubility of each element in the phase. What is the PREN value of each phase? Which phase is undergoing pitting corrosion?

How is the galvanic corrosion process in these materials. Nickel-rich austenite (cathode) - nobler electrode potential in saline environment than Cr and Fe; ferrite - lower electrode potential (anode). Explain in terms of area ratio.

How can the corrosion rate of steel 8 reach a maximum of 65% ferrite and then drop with increasing ferrite fraction? What is the explanation for this behavior.

Do water quenching nitrides degrade the corrosion resistance?

The polarization curves shown in Figure 10 have been little explored. It is necessary to discuss the observed differences in corrosion potentials with respect to the types of elements used for cleaning the DSS. In addition, the Ecorr, Epit values in Table 3 should be expressed with respect to the reference electrode used. The same applies to the potential scan shown in Figure 10 on the y-axis.

Author Response

Thank you for your comments!

Reviewer 3 Report

The present work is devoted to the study of a methodology for quality management of DSS. The work is interesting and well written but some major revision are required before publication.

a)       I suggest to improve the introduction section adding some recent works regarding microstructural and corrosion properties of DSS such as: DOI 10.3390/met8121074 and DOI 10.3390/ma12121911

b)      I suggest to improve the experimental section adding the reproducibility of the tests and the method for preparation of the specimens for each test

c)       I suggest to revise the structure of the results and discussion chapter. In particular I suggest first to present the model and after to present the results that validate the model, without dividing in subchapter for each single property

d)      The microstructural part has to be properly improved. Now are presented only some selected microstructures in Fig.2 but are not explained at what steel they belong. I suggest to add microstructural analysis for all the selected steels in order to confirm the obtained results.

e)      Regarding the presentation of Fig.10 I suggest to add proper arrows to indicate pitting potential passive zone etc

f)        In general the main lack of the work is the structure of the results and discussion part that is a bit confused, the authors perform very interesting work that is not presented clearly. So I suggest to present first the result of the proposed model and after the validation of model with the experiments dividing all the presentation in two sub chapters and not dividing on the base of the selected property.

g)       I suggest to improve also the discussion section on the base of the results of the present literature

h)      I suggest to write in a more concise way the conclusion section

Author Response

Thank you for your comments!

Round 2

Reviewer 1 Report

The paper can be accepted in its present form.

Author Response

Thanks again for your comments!

Reviewer 2 Report

The authors improve the manuscript, but there are still minor revisions that should be made.

On the y-axis in Figure 10, please add the reference electrode, for example: potential V(Ag/AgCl). This must also be mentioned in Table 3 for Ecorr and Epit. In addition, the authors must add the experimental error of the measurements.

Author Response

Thank you for your comments! We added the reference electrode on the y-axis in Fig. 10 and in Table 3 for Ecorr and Epit.

The experimental error of potentiodynamic tests is determined by the passport error of the instruments.

Reviewer 3 Report

I suggest publication in its form

Author Response

Thanks again for your comments!